# Fat tissue regulates the pathogenesis and severity of cardiomyopathy in murine chagas disease

Kezia Lizardo[1], Janeesh P. Ayyappan[2], Neelam Oswal[1], Louis M. Weiss🄳[3,4], Philipp E. Scherer🄳[5], Jyothi F. Nagajyothi🄳[1] *

**1** Center for Discovery and Innovation, Hackensack University Medical Center, Nutley, New Jersey, United States of America, **2** Department of Biochemisty, University of Kerala, Kerala, India, **3** Department of Pathology, Albert Einstein College of Medicine, Bronx, New York, New York, United States of America, **4** Department of Medicine, Albert Einstein College of medicine, Bronx, New York, New York, United States of America, **5** The Touchstone Diabetes Center, UT Southwestern Medical Center, Dallas, Texas, United States of America

* Jyothi.Nagajyothi@HMH-CDI.org

**Data Availability Statement:** All relevant data are within the manuscript and its Supporting Information files.

## Abstract

Chronic Chagas cardiomyopathy (CCC) caused by a parasite *Trypanosoma cruzi* is a life-threatening disease in Latin America, for which there is no effective drug or vaccine. The pathogenesis of CCC is complex and multifactorial. Previously, we demonstrated *T. cruzi* infected mice lose a significant amount of fat tissue which correlates with progression of CCC. Based on this an investigation was undertaken during both acute and chronic *T. cruzi* infection utilizing the FAT-ATTAC murine model (that allows modulation of fat mass) to understand the consequences of the loss of adipocytes in the regulation of cardiac parasite load, parasite persistence, inflammation, mitochondrial stress, ER stress, survival, CCC progression and CCC severity. Mice were infected intraperitoneally with 5x10⁴ and 10³ try-pomastigotes to generate acute and chronic Chagas models, respectively. Ablation of adipocytes was carried out in uninfected and infected mice by treatment with AP21087 for 10 days starting at 15DPI (acute infection) and at 65DPI (indeterminate infection). During acute infection, cardiac ultrasound imaging, histological, and biochemical analyses demonstrated that fat ablation increased cardiac parasite load, cardiac pathology and right ventricular dilation and decreased survival. During chronic indeterminate infection ablation of fat cells increased cardiac pathology and caused bi-ventricular dilation. These data demonstrate that dysfunctional adipose tissue not only affects cardiac metabolism but also the inflammatory status, morphology and physiology of the myocardium and increases the risk of progression and severity of CCC in murine Chagas disease.

## Author summary

An estimated eight million individuals worldwide are chronically infected with *Trypanosoma cruzi*, the causative agent of Chagas disease (CD). Of these infected individuals, 30% will develop chronic Chagas cardiomyopathy (CCC), a major cause of morbidity and

**Funding:** This study was supported by grants from the National Heart, Lung, and Blood Institute (National Institutes of Health HL-122866) to JNJ. The funders had no role in study design, data collection and analysis, decision to publish, or preparation of the manuscript.

**Competing interests:** No authors have competing interests.

mortality in CD endemic regions for which there is currently no effective drug or vaccine. The molecular mechanisms underlying CCC pathogenesis, progression and severity are complex, multi-factorial and not completely understood. Earlier, it was demonstrated that *T. cruzi* persists in adipose tissue, alters adipocyte physiology, and causes loss of body fat mass in *T. cruzi* infected mice with CCC. In this study, the authors examined the role of visceral fat pad (adipose tissue) in regulating the pathogenic signalling in the development and progression of CCC using a fat mass modulatable transgenic mouse CD model. Loss of fat cells increased cardiac lipid load and deregulated cardiac lipid metabolism leading to mitochondrial oxidative stress and endoplasmic reticulum stress and severe CCC. In addition, loss of fat cells increased cardiac parasite load during acute infection and altered immune signalling in the hearts of infected mice during chronic infection. These discoveries underscore the importance of adipose tissue in the development of CCC.

## Introduction

Chronic Chagas cardiomyopathy (CCC) is a devastating heart disease caused by infection with *Trypanosoma cruzi*. Following initial infection cardiomyopathy can develop in approximately 30% of patients after a prolonged duration of time. The disease severity varies in patients with chronic Chagas Disease (CD) and the spectrum of CCC includes EKG abnormalities, myocarditis, cardiac hypertrophy, progressive myocardial fibrosis, left ventricular dilatation and dysfunction, ventricular aneurysm, congestive heart failure, thromboembolism, ventricular arrhythmias, cardiac conduction system abnormalities and sudden cardiac death. CCC is a leading cause of death related to cardiovascular diseases in the CD endemic regions of Latin America. Globalization has increased the cases of patients with CCC/CD in developed countries due to immigration. Currently, CD affects 6 to 8 million people globally.

Despite this disease being described for over 100 years, the cellular and molecular mechanisms contributing and driving the various disease manifestations, e.g. ventricular hypertrophy to severe multi-ventricular dilation, are still not completely understood. Following infection, the development of acute myocarditis is attributed mainly to cardiac parasite load and pro-inflammatory signaling. However, the cellular mechanisms involved in the pathogenesis of the clinical manifestations of chronic CD are multifactorial and while the persistence of parasites and the presence of inflammatory signaling, which varies among patients, is involved, other mechanisms such as inflammation-induced cell death followed by fibrosis are important in the development of CCC.

Lipids and lipotoxicity have now been recognized to have a role in various heart diseases [1,2]. Previously, our research group demonstrated that cardiac lipidopathy leading to endoplasmic reticulum (ER) stress and mitochondrial oxidative stress is a factor in the development of CCC [3]. *T. cruzi* infected mice that are treated with the ER stress inhibitor, 2-Aminopurine or the lipid biosynthesis inhibitor Betulin have a significant modulation in cardiomyopathy, mitochondrial stress, and ER stress during chronic infection [4]. Cardiac lipid levels are regulated by many intrinsic and external factors including genetics, diet, and metabolic status [5]. High-fat diet differentially regulates cardiac parasite load, lipid accumulation, and pathology and survival rate during acute and chronic *T. cruzi* infection in mice and this effect is dependent on the observed body fat mass [6].

Adipose tissue (fat tissue) and the liver play important roles in maintaining and regulating cardiac lipid metabolism. *T. cruzi* persists in adipose tissue and alters the functions of adipose tissue by regulating lipolysis and inflammation [7–9]. Cardiac fat tissue (epicardial and pericardial fat tissue) is located in proximity to heart muscle and probably plays an important role

in regulating cardiac parasite load, inflammation, and metabolic functions [10]. Clinical studies have shown an association between body mass index (BMI), fat mass, dyslipidemia, nutritional status, and CCC severity [11,12]. However, it is not known whether adipocytes directly regulate the pathogenesis of CCC. In the current study, our laboratory group investigated the role of altered adipocyte levels and physiology, using a fat-amendable transgenic murine FAT-ATTAC (fat apoptosis through targeted activation of caspase 8) model, on the regulation of cardiac parasite load, parasite persistence, inflammation, mitochondrial stress, ER stress, and CCC progression and severity, and survival during acute and chronic *T. cruzi* infection [13–15]. FAT-ATTAC mice are indistinguishable from their wild-type littermates; however, apoptosis of adipocytes can be induced at any developmental stage by administration of an FK1012 analog, leading to the dimerization of a membrane-bound, adipocyte-specific caspase 8-FK506 binding protein (FKBP) fusion. Within 2 weeks of the dimerizer administration, FAT-ATTAC mice have severely reduced (95%) levels of circulating adipokines and substantially reduced levels of adipose tissue [13,14]. This model has allowed us to directly test the role of adipocytes and adipose tissue physiology in regulating cardiac pathology, parasite load burden, and immune status by selectively ablating fat tissue during acute and indeterminate stages of *T. cruzi* infection in a murine model. Ablation of fat cells increased cardiac parasite load, cardiac pathology and right ventricular dilation and decreases survival in the infected mice during acute infection. To investigate the effect of loss of fat cells in the transition from the indeterminate stage to the determinate chronic stage, fat ablation was induced in the infected mice at the early stage of chronic infection. Ablation of fat cells increased cardiac pathology and caused bi-ventricular dilation in infected mice during early chronic stages of infection. Loss of adipocytes contributed to cardiac lipidopathy and associated mitochondrial and ER stress and increased the risk of a severe form of CCC. These results suggest that adipocytes participate in the pathogenesis and progression of CCC via multiple signaling pathways.

## Materials and methods

All experimental animal protocols were approved by the Institutional Animal Care and Use and Institutional Biosafety Committees of Hackensack University Medical Center (IACUC #282) and Rutgers University (IACUC# PROTO999900866) and adhere to the National Research Council guidelines.

### Animal model and experimental design

The Brazil strain of *T cruzi* (DTU1, 21) was maintained by passage in C3H/Hej mice (Jackson Laboratories, Bar Harbor, ME) [16]. The transgenic FAT-ATTAC mice (generous gift of Dr. Scherer, Texas) were bred at New Jersey Medical School, Rutgers University. Mice were maintained on a 12-h light, 12-h dark cycle and housed in groups of two to four, with unlimited access to water and chow (no. 5058; LabDiet). 6- to 7-wk-old (both male and female) FAT-ATTAC mice were infected with trypomastigotes of the Brazil strain to generate the murine models of acute and early chronic Chagas disease (CD) as described below (S1 Fig).

### Acute CD model

Mice (n = 20) were infected intraperitoneally with $5x10^4$ trypomastigotes. A separate group of uninfected mice (n = 20) were included as controls. At 15 days post infection (DPI), one set of infected and uninfected mice (n = 10/group) were administered AP21087 (see below) to induce fat ablation. Mice were sacrificed 30 DPI and hearts and visceral fat pads (epididymal fat pads) were harvested. The experiment was performed twice using the same number of mice in each group.

### Indeterminate/early chronic CD model

Mice (n = 25, expecting 35% mortality during acute stage) were infected intraperitoneally with $10^3$ trypomastigotes. A separate group of uninfected mice (n = 20) were included as controls. At 65 DPI, one set of infected and uninfected mice (n = 10/group) were administered AP21087 (see below) to induce fat ablation. Mice were sacrificed 90 DPI and hearts and visceral fat pads (epididymal fat pads) were harvested. Portions of the harvested tissues were fixed in 10% formalin for histological analysis. Portions of tissues were also stored immediately at −80˚C for total RNA isolation and protein extraction. A flow chart describing the experimental design is presented as S1 Fig.

### Protocol for fat ablation

AP21087 ((B/B homodimerizer, Takara Bio, CA, USA) was administered by intra-peritoneal (ip) injection at a dose of 0.5 μg/g of body weight once daily for 10 d starting at 15DPI (acute CD model) and at 65DPI (indeterminate CD model).

### Cardiac ultrasound analysis

Cardiac ultrasound imaging of the mice was performed at 30 DPI (acute stage of infection) and 90 DPI (early chronic stage of infection) using a Vevo 2100 ultra-high-frequency ultrasound system (Visual Sonics Inc, Toronto, Canada) at the Rutgers University Molecular Imaging Center, as previously published [3]. B-mode, M-mode, and pulse wave Doppler image files were collected from both the parasternal long-axis and short-axis views. Morphometric measurements were determined using image analysis tools available in the Vevo workstation software.

### Histological analyses

Freshly isolated tissues were fixed with phosphate-buffered formalin for a minimum of 48 hours and then embedded in paraffin wax. Hematoxylin and eosin and Masson chrome staining were performed, and the images were captured as previously published [7]. Four to six images/section (4x magnified) of each tissue were scored blindly. For adipose tissue samples, histologic evidence of pathology was classified in terms of infiltrated immune cells, degraded lipid droplets (micro lipid droplets (≤20um) and macro lipid droplets (≥50um) and dead adipocytes (without lipid droplets) on a six-point scale ranging from 0 to 5+. For each heart sample, histologic evidence of pathology was classified in terms of infiltrated immune cells, lipid droplets (macro lipid (≥20um) and micro lipid (≤20um)), and presence of amastigote nests and was graded on a six-point scale ranging from 0 to 5+. For each fat tissue sample, histologic evidence of pathology was classified in terms of infiltrated immune cells and size of lipid droplets was graded on a six-point scale ranging from 0 to 5+.

### Immunoblot analysis

Protein lysates of the heart and adipose tissue samples were prepared and immunoblot analysis performed as published previously [7]. Adiponectin-specific mouse monoclonal antibody (1:1000 dilution, AB22554, Abcam), peroxisome proliferator-activated receptors (PPAR)-γ specific rabbit monoclonal antibody (1:1000 dilution, C26H12, Cell Signaling), PPAR-α-specific mouse monoclonal antibody (1:1000 dilution, MA1-822, Thermo scientific), Hormone sensitive lipase (HSL)-specific rabbit monoclonal antibody (1:1000 dilution, D6W5S, Cell Signaling), Phospho-HSL (Ser563)-specific rabbit polyclonal antibody (1:1000 dilution, #4139, Cell Signaling), Adipose Triglyceride lipase (ATGL)-specific rabbit monoclonal antibody

(1:1000 dilution, 30A4, Cell Signaling), F4/80-specific mouse monoclonal antibody (1:500 dilution, sc-377009, Santa Cruz Biotechnology, INC), Tumor necrosis factor (TNF)-α-specific rabbit polyclonal antibody (1:2000 dilution, AB6671, Abcam), Fatty acid binding protein (FABP)-4-specific rabbit monoclonal antibody (1:1000 dilution, D25B3, Cell Signaling), BNIP3 specific rabbit monoclonal antibody (1:1000 dilution, #3769, Cell Signaling), Caspase 3 specific polyclonal antibody (1:1000 dilution, #9662, Cell Signaling), Acyl-CoA synthetase (ACSL)-1-specific rabbit monoclonal antibody (1:1000 dilution, #9189, Cell Signaling), CETP-specific mouse monoclonal antibody (1:1000 dilution, ATM192, Abcam), ATP binding cassette transporter (ABCA)-1-specific mouse monoclonal antibody (1:500 dilution, AB.H10, Abcam), Phospho-perilipin1 (Ser522)-specific mouse monoclonal antibody (1:1000 dilution, #4856, Vala Sciences), Cytochrome c-specific rabbit monoclonal antibody (1:1000 dilution, 136F3, Cell Signaling), Anti-prohibitin (PHB)-1-rabbit polyclonal antibody (1:1000 dilution, #2426, Cell Signaling), Superoxide dismutase (SOD)-1-specific mouse monoclonal antibody (1:1000 dilution, 71G8, Cell Signaling), Heat shock protein (HSP)-60-specific rabbit monoclonal antibody (1:1000 dilution, D6F1, Cell Signaling), Succinate dehydrogenase subunit A (SDHA)-specific rabbit monoclonal antibody (1:1000 dilution, D6J9M, Cell Signaling), Pyruvate Dehydrogenase-specific rabbit monoclonal antibody (1:1000 dilution, C54G1, Cell Signaling), CCAAT-enhancer binding protein (CHOP)-specific mouse monoclonal antibody (1:1000 dilution, L63F7, Cell Signaling), Binding immunoglobulin protein (BIP)-specific rabbit monoclonal antibody (1:1000 dilution, C50B12, Cell Signaling), Low density lipoprotein receptor (LDLR)—specific rabbit monoclonal antibody (1:1000 dilution, ab52818, Abcam), Acetyl CoA carboxylase specific rabbit monoclonal antibody (1:1000 dilution, C83B10, Cell Signaling), acetyl-coenzyme A acetyl transferase (ACAT)-1-specific rabbit polyclonal antibody (1:1000 dilution, #44276, Cell Signaling), Carnitine palmitoyltransferase 1A (CPT1A)-specific mouse monoclonal antibody (1:1000 dilution, 8F6AE9, Abcam) and interferon gamma (IFNγ)-specific rabbit monoclonal antibody (1:1000 dilution, EPR1108, Abcam) were used as primary antibodies. Horseradish peroxidase (HRP)-conjugated goat anti-mouse immunoglobulin (1:2000 dilution, Thermo Scientific) or HRP-conjugated goat anti-rabbit immunoglobulin (1:2000 dilution, Thermo Scientific) were used to detect specific protein bands (as noted in the figure legends) using a chemiluminescence system. Guanosine nucleotide dissociation inhibitor (GDI) (1: 10,000 dilution, 71–0300, rabbit polyclonal, Invitrogen, CA) and a secondary HRP-conjugated goat anti-rabbit antibody (1:2000 dilution, Amersham Biosciences) were used to normalize protein loading.

## Quantitative determination of parasite load in tissue

Quantitative PCR was performed using genomic DNA (10ng) from the heart and adipose tissue samples (90DPI), *Trypanosma cruzi* specific primers (TCZ-F and TCZ-R) [16] and SYBR Green master mix (Qiagen) following the manufacturers protocol (Qiagen). Genomic DNA of *T. cruzi* was used to generate a standard curve [16]. Real-time PCR was conducted using the Applied Biosystem Quant Studio-3 system and quantified the parasite load.

## Statistical analysis

For immunoblotting analysis, the densitometric values of the immunoreactive bands (immunoblotting) were analyzed with the Image Studio lite package V5.2 (LI-COR Biosciences, Lincoln, NE). Statistical analyses were performed using a Student t test (Microsoft Excel), as appropriate, to compare between 2 groups. Results were expressed as mean ± SE. Significant differences were reported for p values between <0.05 and <0.001.

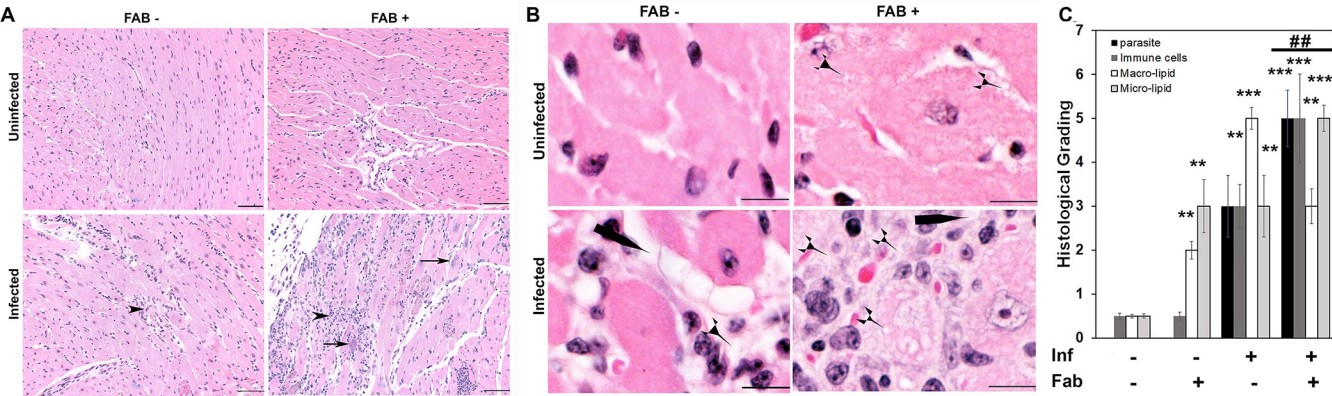

**Fig 1. Fat ablation increases cardiac parasite load, lipid accumulation and cardiac pathology in *T. cruzi* infected acute (30 DPI) CD murine model (n = 6; minimum 5 images/section were analyzed). A.** Hematoxylin and eosin (H&E) staining of hearts in indicated mice (infected or uninfected mice, fat-ablated (Fab +) or fat-unablated (Fab-). Infiltrated immune cells, black arrowhead; amastigote nests, black long arrow, and presence of lipid granules (see S2 Fig). Bar = 100 μm, 20x magnification. Additional images are presented as S3 Fig. **B.**Magnified (40x) images of H&E stained heart sections of indicated mice showing macro-vesicular (black long pointer) and micro-vesicular (black short pointer) lipid droplets. Bar = 50 um. **C.**Histologic grading of heart tissue pathology was carried out according to experimental groups and classified in terms of degree of parasite load, infiltrated immune cells, size of lipid droplets (macro lipid (≥20um) and micro lipid (≤20um) droplets) in the H&E sections of hearts in acute *T. cruzi* infected mice with and without fat ablation. The values plotted are mean ± standard deviation (SD) from n = 5. \*\*, p < 0.01\*\*\*, p < 0.001.

## Results

### Loss in adipocytes affects cardiac pathology and survival rate in *T. cruzi* infected mice during acute infection

Mice infected with $5x10^4$ parasites (Brazil strain) developed acute myocarditis and cardiomyopathy by 25 DPI. The effect of loss of adipocytes on cardiac pathology and survival in infected mice was analysed by ablating fat cells via induced apoptosis for 10 days (between 15 and 25 DPI) [7,17] (S1 Fig). An approximately 80% loss in body fat in fat-ablated mice was observed during the dissection compared to uninfected fat-unablated mice as previously published [15]. Histological analysis demonstrated the presence of amastigote nests and increased infiltration of immune cells in the hearts of *T. cruzi* infected mice (Fig 1A). *T. cruzi* infection increased the accumulation of macro lipid bodies, whereas, the fat ablation increased the accumulation of micro lipid bodies in the hearts of infected mice as demonstrated by histological analysis (Fig 1B and 1C). Fat ablation during acute infection significantly increased parasite load, the number of infiltrated immune cells and enhanced cell damage compared with infected fat-unablated (infected) mice as demonstrated by the histologic analysis (Fig 1A and 1C). Fat-ablation increased the mortality of *T. cruzi* infected mice resulting in a 50% survival rate compared with the 65% survival rate in the infected mice that had not been ablated (S2A Fig).

Cardiac ultrasound imaging analysis was performed to evaluate the effect of fat ablation on cardiac morphology during acute infection (Fig 2). This morphological analysis demonstrated no significant change in the left ventricle internal diameter diastole (LVIDd) in mice with and without infection and with and without fat ablation. The LVID systole was reduced (p<0.05) in the infected fat-ablated mice compared with the mice in other groups. The right ventricle internal diameter (RVID) both diastole and systole increased (p<0.05 and p<0.05, respectively) in the infected mice compared with uninfected fat un-ablated (control mice). Fat ablation in the infected mice increased RVID diastole and RVID systole compared to control *T. cruzi* infected mice (1.5-fold, p<0.05 and 1.5-fold, p<0.05) as well compared to the uninfected control (3-fold, p<0.01 and 2.5-fold, p<0.001) mice. Fat ablation increased RVID in

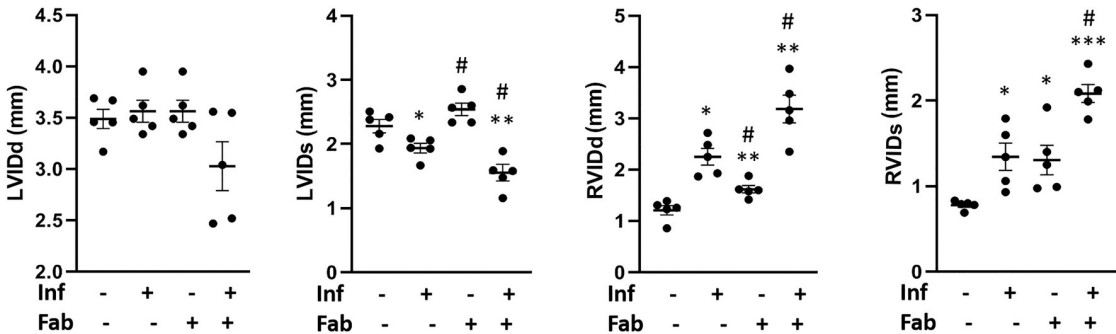

**Fig 2. Loss in adipocytes increases cardiac enlargement during acute *T. cruzi* infection (30DPI).** Left ventricle internal diameter (LVID) and right ventricle internal diameter (RVID), measured by ultrasound analysis of the hearts both at diastole (d) and systole (s) conditions at 30 DPI, in infected or uninfected mice, fat-ablated (Fab +) or fat-unablated (Fab-) mice as indicated. The error bars represent SEM. *$p \leq 0.05$, **$p \leq 0.01$ or ***$p \leq 0.001$ compared with uninfected fat-unablated. #$p \leq 0.05$ compared with infected fat-unablated.

uninfected mice compared with fat uninfected unablated (control) mice. Doppler flow profiles demonstrated a significant difference ($p \leq 0.01$) in the ejection fractions (%) between uninfected mice (EF%- 69 ± 5), uninfected fat-ablated mice (EF%- 55 ± 4.7), and infected fat-ablated mice (EF%- 81 ± 6.1). We did not detect any significant difference in EF% between uninfected and infected mice. The fractional shortening (FS) measurements demonstrated a significant difference ($p \leq 0.01$) in the FS% between uninfected mice and fat-ablated mice (uninfected and infected) following similar trends to that seen in EF%. Thus, fat ablation during acute infection caused increased cardiac lipid accumulation, increased infiltration of immune cells, and increased parasite load, increased cardiomyopathy and increased mortality in *T. cruzi* infected mice.

## *T. cruzi* infection alters adipocyte morphology and adipose tissue physiology during the chronic stage of infection

Mice infected with $10^3$ T. cruzi (Brazil strain) generally display cardiomyopathy around 90 DPI, which we define as the early chronic phase of infection. We analyzed the effect of fat ablation during the indeterminate stage (65 DPI) on the pathogenesis and progression of cardiomyopathy at 90DPI (early chronic infection). Histologic analysis of adipose tissue demonstrated that adipocytes in infected mice varied in size ranging from 5 to 130um compared with the uniformly distributed adipocytes (size ranging between 50 -100um) in control mice (Fig 3A–3B). Histologic analysis demonstrated increased levels of infiltrated immune cells (p value) in the adipose tissues of infected mice compared with uninfected mice (Fig 3A–3B). Some of the adipocytes had multi-ocular lipid droplets and were surrounded by infiltrating immune cells (Fig 3A). Fat ablation via induced apoptosis further increased the number of dead adipocytes and infiltrated immune cells seen by histologic analysis (Fig 3A and 3B). Parasite persistence in adipose tissue and if fat-ablation affects parasite load in adipose tissue during chronic infection was analyzed using qPCR [16]. Quantitative PCR analysis showed significantly reduced amount of parasite DNA in adipose tissue in infected fat-ablated mice compared to infected mice (S3A Fig).

The effect of infection and fat-ablation on adipogenesis in adipose tissue was then analysed. Immunoblot analyses of fat tissues demonstrated significant alterations in the protein levels of adipogenic markers, such as adiponectin, FABP4 and PPARγ in *T.cruzi*-infected mice compared with control mice (Fig 4). The levels of adiponectin were reduced (p<0.05), FABP4 levels were increased (p<0.01) and PPARγ levels were not changed during infection compared to

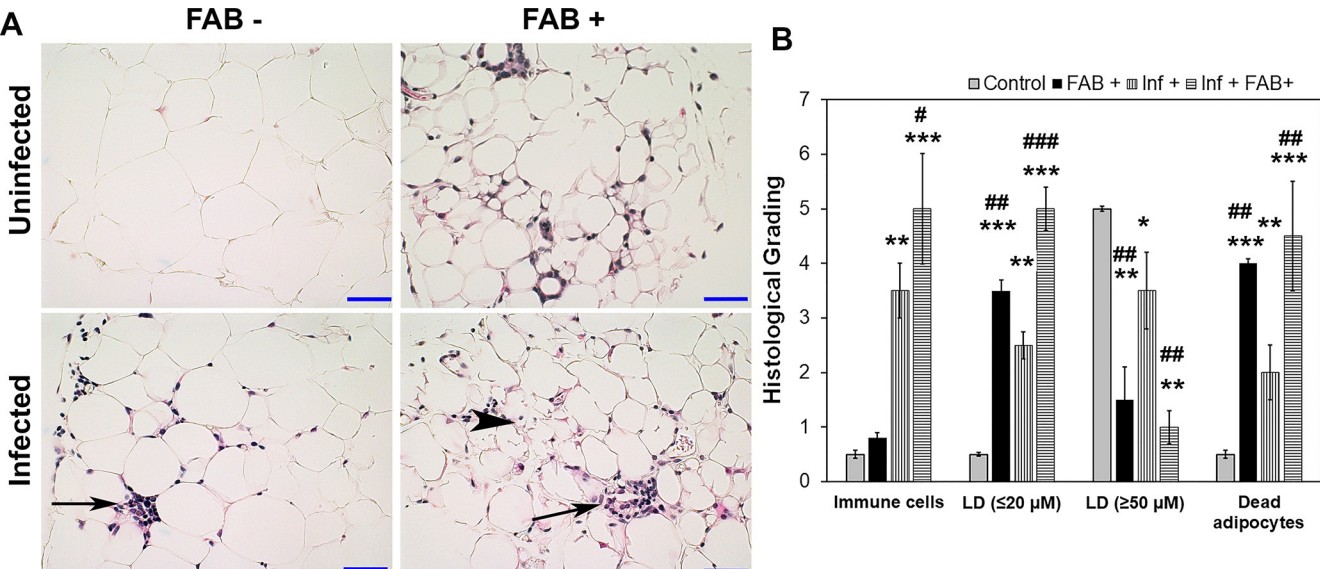

**Fig 3. *T. cruzi* infection alters adipose tissue pathology and fat-ablation further increases infiltration of immune cells into adipose tissue in FAT-ATTAC mice during early chronic stage (90 DPI). A.** Hematoxylin and eosin (H&E) staining of adipose tissue in indicated mice (infected or uninfected mice, fat-ablated (Fab +) or fat-unablated (Fab-) n = 5). Infiltration of immune cells, long black arrow; multi-ocular lipid droplets/smaller size lipid droplets, black arrowhead. Bar = 50 μm, 20x magnification. **B.** Histological grading of adipose tissue pathology was carried out according to experimental groups and classified in terms of degree of infiltrated immune cells, size of adipocytes (micro-lipid droplets ≤20 um, and macro-lipid droplets ≥50 um), and number of dead adipocytes in adipose tissue during chronic *T. cruzi* infection and/or fat ablation (5 images per section/mouse in each group. Each class was graded on a six-point scale ranging from 0 to 5+ as discussed in Method section and presented as a bar graph. The values plotted are mean ± standard deviation (SD) from n = 5. *p≤0.05, **p≤0.01 or ***p≤0.001 compared with uninfected fat-unablated. #p≤0.05, ##p≤0.01 or ###p≤0.001 compared with infected fat-unablated.

the levels in control mice (Fig 4A). However, the levels of adiponectin, FABP4 and PPARγ were all increased in the fat tissues of infected fat-ablated mice compared with infected mice (11-fold p<0.001, 1.5-fold p<0.01, and 1.4-fold p<0.01, respectively), as well as compared to control mice (5-fold p<0.01, 7-fold p<0.001, and 1.7-fold p<0.01, respectively) (Fig 4A).

We analyzed the effect of infection and fat-ablation on lipolysis in adipose tissue (Fig 4B). Immunoblot analysis demonstrated a reduction in the levels of lipases such as ATGL, HSL and p-HSL (1.7-fold p<0.01, 2.5-fold, p<0.001 and 1.2-fold, p<0.01) in the fat tissues of infected mice as compared to control mice (Fig 4B). Fat ablation increased the levels of ATGL (1.7-fold, p<0.01) in infected mice compared with infected fat-unablated mice and infected mice did not show any significant difference compared to control mice. Interestingly, the levels of p-perilipin and PPARα (2.33-fold p<0.05, and 1.7-fold and p<0.001, respectively) were increased in the adipose tissue of infected mice compared with control mice as demonstrated by immunoblot analysis (Fig 4B). Fat-ablation in infected mice increased the levels of p-perilipin and PPARα (1.4-fold p<0.001, and 2.2-fold and p<0.001, respectively) compared with infected mice and also compared to control mice (4-fold p<0.001, and 3.8-fold and p<0.001, respectively).

These results suggest that persistence of parasites and infiltrated immune cells in adipose tissue alters adipose tissue physiology by causing an imbalance between adipogenesis and lipolysis.

## Chronic *T. cruzi* infection induces inflammation and cell death in adipose tissue via necrosis and apoptosis

The effect of the loss of adipocytes and increased lipolysis on the infiltration of immune cells and inflammatory signaling into adipose tissue during infection and fat-ablation was evaluated

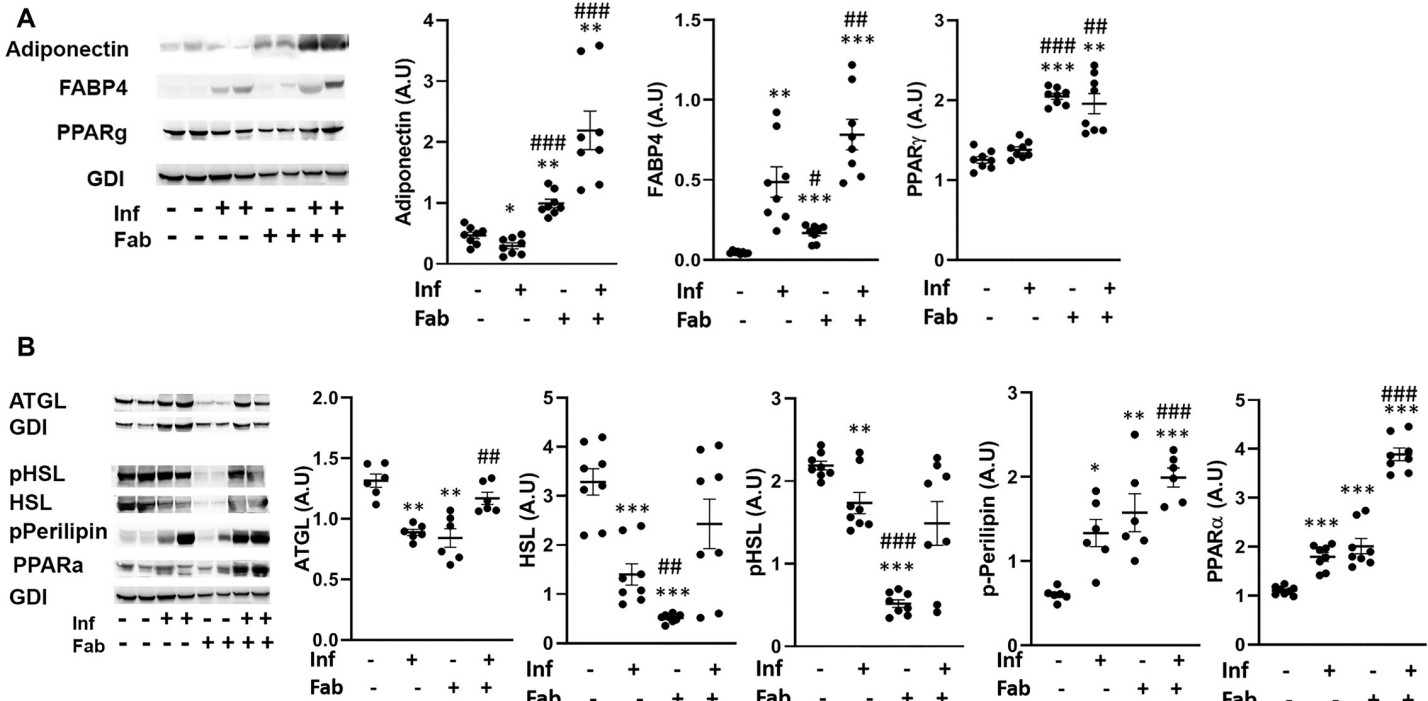

**Fig 4. *T. cruzi* infection and fat ablation alter adipogenesis and lipolysis in adipose tissue during chronic stage in the infected mice. A.** Immunoblot analyses of (A) adipogenic markers (adiponectin, FABP4 and PPAR) and (B) lipid degradation markers such as lipases (ATGL, HSL and p-HSL), loss of lipid droplets (p-perilipin) and lipid oxidation (PPARα) in adipose tissue of indicated mice (infected or uninfected mice, fat-ablated (Fab +) or fat-unablated (Fab-) n = 8). The change in protein levels were normalized to the levels of Guanosine nucleotide dissociation inhibitor (GDI) and plotted column scatter graph. The error bars represent SEM. A.U. indicates arbitrary unit. *p≤0.05, **p≤0.01 or ***p≤0.001 compared with uninfected fat-unablated. #p≤0.05, ##p≤0.01 or ###p≤0.001 compared with infected fat-unablated.

(Fig 5). Immunoblot analysis demonstrated increased levels of infiltrated macrophages as indicated by F4/80 levels in the adipose tissue of infected mice (6.2-fold, p<0.01) and fat ablation in infected mice further increased the levels of macrophages (12.5-fold, p<0.01), compared to control mice (Fig 5A). Although, the levels of macrophages were increased in infected mice, the levels of INFγ and TNFα did not increase and TNFα was reduced (1.2- fold, p<0.01) compared to control mice (Fig 5A). However, fat-ablation in infected mice increased the levels of TNFα (1.2-fold, p<0.05) compared to infected mice (Fig 5A). Immunoblot analysis showed a significant increase in the levels of the necrosis marker BNIP3 (1.8-fold, p<0.001) in the adipose tissue of *T. cruzi*-infected mice as compared to control mice (Fig 5B). This further increased in infected fat-ablated mice as compared to both infected and control mice (1.8-fold, p<0.001 and 4-fold, p<0.001, respectively) (Fig 5B).

There was an increase in apoptosis as determined by the apoptosis marker, cleaved Caspase3 (6-fold, p<0.01), with chronic *T. cruzi* infection as compared to uninfected mice (Fig 5B). As expected, fat ablation further increased both cleaved caspase3 (2-fold, p<0.001 and 12-fold, p<0.001 respectively) and caspase 3 (2-fold, p<0.001 and 2-fold, p<0.001 respectively) in infected mice as compared with infected fat-unablated and control mice (Fig 5B).

These data indicate that the presence of dysfunctional adipocytes and persistence of parasites increase infiltration of immune cells and induce adipocyte cell death via necrosis and apoptosis in adipose tissue during the indeterminate/early chronic infection in *T. cruzi* infected mice.

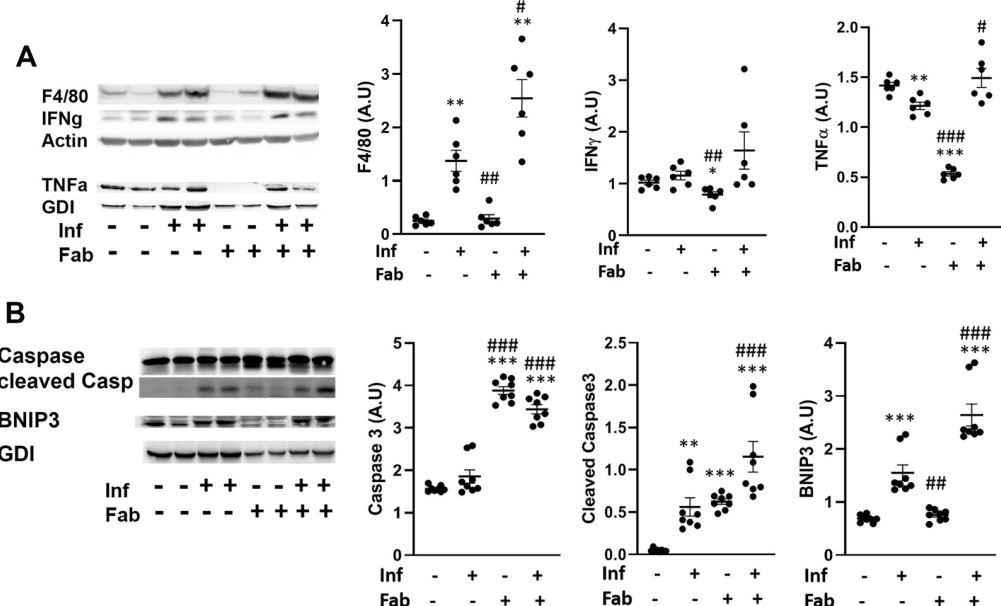

**Fig 5. *T. cruzi* infection induces inflammation and cell death in adipose tissue via necrosis and apoptosis.** A. Immunoblot analysis of (A) markers of inflammation (infiltrated macrophages (F4/80), and cytokines (INFγ and TNFα)), and (B) cell death (BNIP3 (a marker of necrosis), and cleaved Caspase3 (a marker of apoptosis)) in the adipose tissue of indicated mice (infected or uninfected mice, fat-ablated (Fab +) or fat-unablated (Fab -) n = 8). The change in protein levels were normalized to the levels of Guanosine nucleotide dissociation inhibitor (GDI) and plotted column scatter graph. The error bars represent SEM. A.U. indicates arbitrary unit. *p≤0.05, **p≤0.01 or ***p≤0.001 compared with uninfected fat-unablated. #p≤0.05, ##p≤0.01 or ###p≤0.001 compared with infected fat-unablated.

## Fat ablation increases adipogenic signaling and elevates lipid levels in the hearts during the early chronic stage in *T. cruzi* infected mice

The effect of *T.cruzi* infection on cardiac adipogenesis, lipid levels and metabolism during the early chronic stage (90DPI) was evaluated in our murine CD model (Fig 6). Immunoblot analysis demonstrated an increase in adipogenic markers, such as FABP4 (2-fold, p<0.05), PPARγ (2-fold, p<0.001) and adiponectin (3-fold, p<0.001) in the hearts of infected mice as compared to control mice (Fig 6A). Fat ablation further increased the levels of FABP4 (2-fold, p<0.01 and 2-fold, p<0.001 respectively), and PPARγ (2-fold, p<0.01 and 5-fold, p<0.001) in the hearts of infected mice as compared with both infected and control mice (Fig 6A).

Immunoblot analysis demonstrated an increase in the cardiac LDL levels in infected mice, though it was not significant compared with control mice at 90DPI (Fig 6B). However, fat-ablation significantly increased the levels of LDL in the hearts of infected mice compared with control mice (1.5-fold, p<0.01) (Fig 6B). Although, the levels of LDL were not significantly increased in the hearts of infected mice at 90DPI, the levels of p-perilipin and PPARα were significantly increased (3-fold, p<0.001 and 2.5-fold, p<0.001) compared with control mice, suggesting that the infiltrated lipids might have been degraded and oxidized at this time point (Fig 6B). Fat-ablation in infected mice further increased the levels of PPARα (1.1-fold, p<0.001) compared with infected mice and also with control mice (3-fold, p<0.001) (Fig 6B). Increased activation of p-perilipin and PPARα should increase the levels of enzymes involved in lipid catabolism and mitochondrial β-oxidation. However, we observed reduced levels of long chain fatty acyl-CoA ligase (ACSL) and acetyl Co-A carboxylase (1.5-fold, p<0.01 and 1.5-fold, p<0.01) in the hearts of infected mice compared with control mice. Interestingly, the levels of ABCA1, which regulates the intracellular cholesterol efflux and CETP (involved in cholesterol

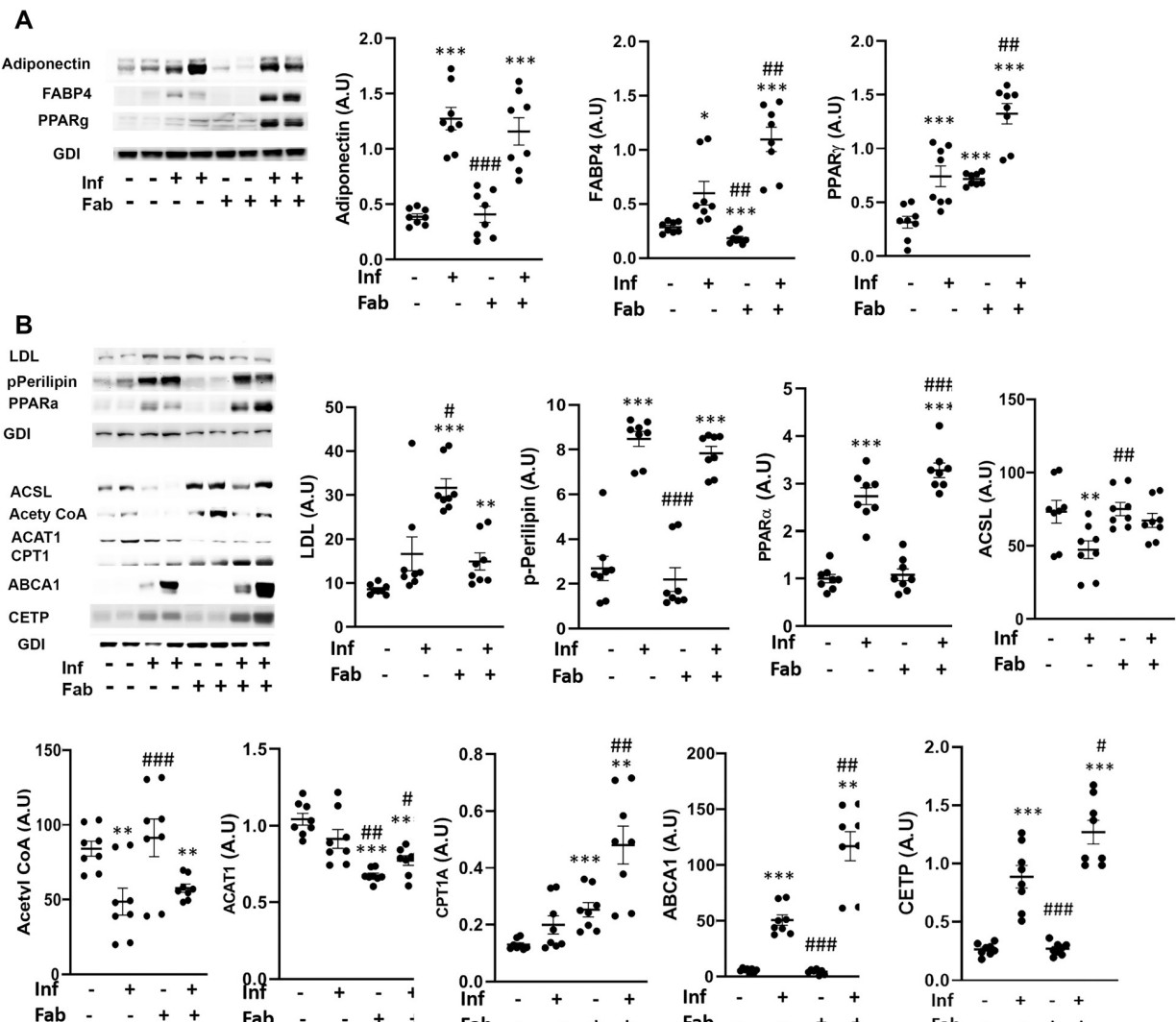

**Fig 6. Fat ablation increases adipogenic signaling and elevates lipid levels in the hearts during the early chronic stage in *T. cruzi* infected mice.** Immunoblot analysis of (A) adipogenic markers (FABP4, PPARγ and adiponectin), and (B) lipid metabolism (LDL and lipid metabolism markers (p-perilipin, PPARα, acyl-CoA ligase (ACSL) and acetyl Co-A carboxylase, CPT1, ABCA1 and CETP) in the hearts of indicated mice (infected or uninfected mice, fat-ablated (Fab +) or fat-unablated (Fab -) n = 8). The change in protein levels were normalized to the levels of Guanosine nucleotide dissociation inhibitor (GDI) and plotted column scatter graph. The error bars represent SEM. A.U. indicates arbitrary unit. *p≤0.05, **p≤0.01 or ***p≤0.001 compared with uninfected fat-unablated. #p≤0.05, ##p≤0.01 or ###p≤0.001 compared with infected fat-unablated.

esterification) were increased in the hearts of infected mice (50-fold, p<0.001 and 3.2-fold, p<0.001) compared with control mice (Fig 6B). Fat-ablation in infected mice did not increase the levels of proteins involved in fatty acid catabolism except for CPT1 (2-fold, p<0.01), however, it increased the levels of ABCA1 and CETP (2.4-fold, p<0.01 and 1.5-fold, p<0.05) compared with infected fat-unablated mice.

## Fat ablation increases mitochondrial dysfunction, ER stress and inflammation in the hearts of chronically infected mice

The increase in cardiac lipid levels leads to over activation of mitochondria and can cause mitochondrial stress resulting to elevation of ER stress. The cardiac level of proteins involved

in mitochondrial function were evaluated including cytochrome C, pyruvate dehydrogenase, heat shock protein 60 (HSP60), superoxide dismutase 1 (SOD1), Prohibitin (PHB1), succinate dehydrogenase (SDHA1) (Fig 7A). We observed a decrease in cytochrome C (1.33-fold and $p < 0.05$), pyruvate dehydrogenase (1.25-fold and $p < 0.01$) and SDHA (1.1-fold and $p < 0.05$) in the hearts of *T.cruzi* infected mice compared to control mice (Fig 7A). The levels of cytochrome C, pyruvate dehydrogenase and SDHA did not significantly change in the hearts between the infected fat-ablated mice and infected mice. The levels of HSP60 and SOD1 increased (1.1-fold, $p < 0.001$ and 2.6-fold, $p < 0.001$, respectively) in the hearts of infected mice compared to control mice (Fig 7A). Fat ablation further elevated the levels of HSP60 and SOD1 in the infected mice (1.2-fold, $p < 0.05$ and 1.2-fold, $p < 0.05$ respectively) compared with infected mice as well as control mice (1.4-fold, $p < 0.01$ and 3-fold, $p < 0.001$) (Fig 7A).

Immunoblot analysis demonstrated an increase in the levels of ER stress markers, BIP (2-fold, $p < 0.01$) and CHOP (9-fold, $p < 0.01$), in *T. cruzi* infected heart tissue as compared to control mice (Fig 7B). The levels of CHOP further increased in the hearts of infected fat-ablated mice as compared with both infected (1.5-fold, $p < 0.05$) and control (6-fold, $p < 0.01$) mice (Fig 7B). Increased mitochondrial stress and ER stress are associated with infiltration of immune cells in the hearts of infected mice, as demonstrated by an increase in the levels of the macrophage marker F4/80 (4-fold, $p < 0.001$) compared to amount seen in control mice (Fig 7C). However, the levels of pro-inflammatory markers such as TNFα and IFNγ demonstrated a significant decrease and no significant change respectively, in the hearts of infected mice compared to control mice (Fig 7C). Interestingly, the levels of TNFα (1.25-fold, $p < 0.01$) further decreased and IFNγ increased (2-fold, $p < 0.001$) in the heart of infected fat-ablated mice compared to infected mice (Fig 7C).

These data demonstrated that loss in fat cells during the indeterminate stage led to significant effects on mitochondrial function, increased ER stress and increased infiltration of IFNγ-producing immune cells in the hearts of *T. cruzi* infected mice during the early stage of chronic infection (90DPI).

## Loss in fat cells exacerbates cardiac pathology and causes bi-ventricular enlargement in the hearts of chronically infected mice

Histologic analysis of myocardium demonstrated significant changes in the morphology seen in heart sections of infected mice compared to that seen in control mice. Analysis of stained tissue sections of hearts demonstrated an absence of amastigote nests in chronically infected mice (Fig 8A). However, qPCR demonstrated the presence of *T. cruzi* specific DNA in the hearts of infected mice, which was significantly higher ($p \leq 0.05$) in infected fat-ablated mice compared to fat un-ablated mice (S3B Fig). Histological staining of infected mice compared to control mice demonstrated the presence of infiltrated immune cells, fibrosis, increased accumulation of macro-lipid droplets, vasculitis, hemorrhage, and a wide separation of the cardiac muscle fibers with an increase in the inter-fiber spaces in the heart sections (Fig 8A and 8B). Fat ablation increased hyaline degeneration, disruption and fragmentation of myofibril striations and further significantly elevated cardiac damage, and accumulation of micro-lipid droplets (Fig 8A and 8B). Masson trichrome–stained sections demonstrated increased fibrosis in the myocardium of the infected mice compared with control mice (S3C Fig). However, no significant change in the levels of fibrosis was observed between the infected fat-ablated and infected fat-unablated mice (S3C Fig).

Cardiac ultrasound imaging at 90DPI demonstrated a slight increase (no significant change) in LVID (both systole and diastole) in infected mice compared to control mice (LVIDs has $p < 0.05$) (Fig 8C). In contrast, RVID of these hearts was significantly increased

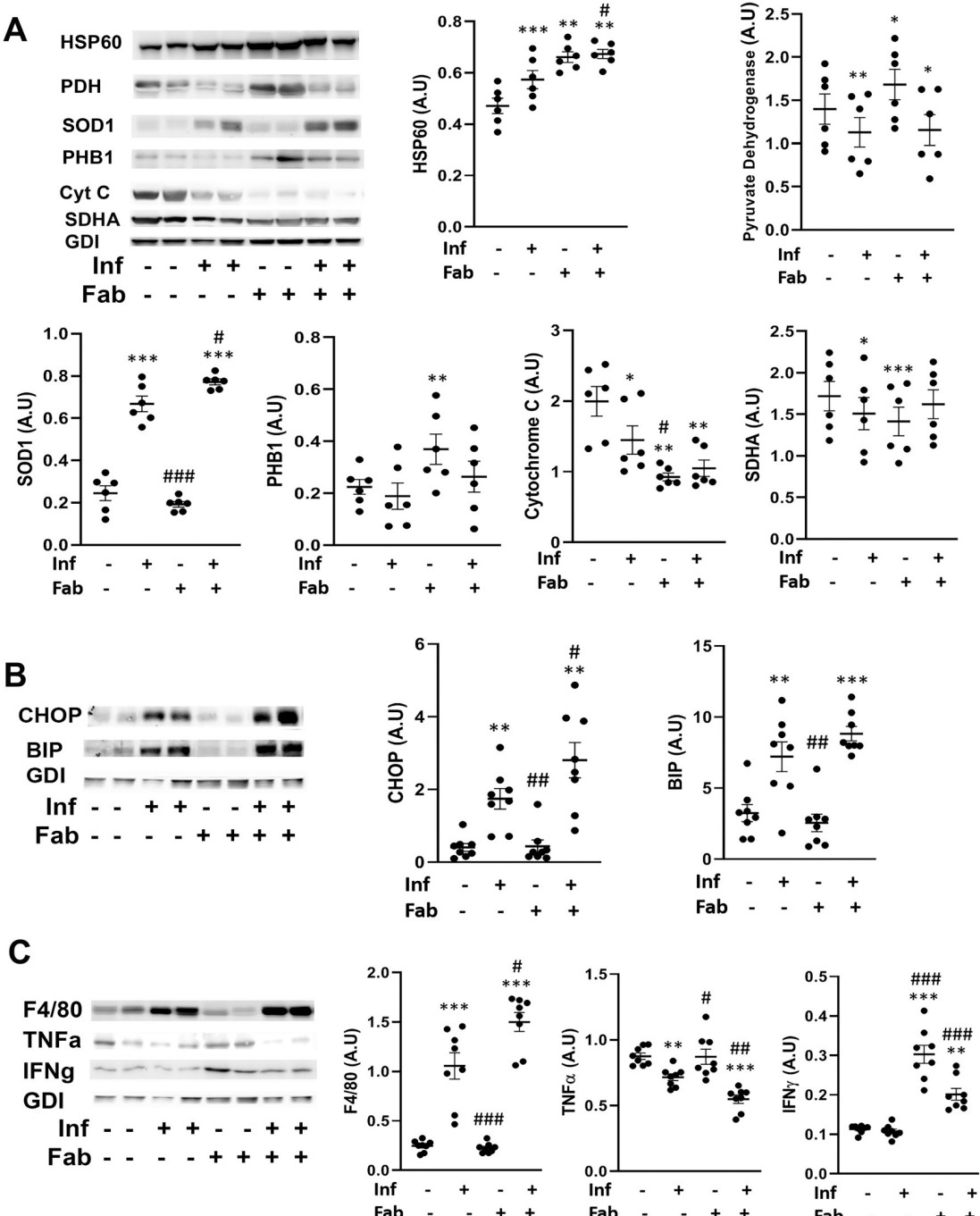

**Fig 7. Fat ablation on mitochondrial dysfunction, ER stress and inflammation in the hearts of chronic CD mice.** A. Immunoblot analysis of (A) markers of mitochondrial function (cytochrome C, pyruvate dehydrogenase, SDHA, HSP60 and SOD1), (B) markers of ER stress (BIP and CHOP) and (C) markers of infiltration of macrophage (F4/80) and cytokines (TNFα and IFNγ) in the hearts of indicated mice (infected or uninfected mice, fat-ablated (Fab +) or fat-unablated (Fab -) n = 8). The change in protein levels were normalized to the levels of Guanosine nucleotide dissociation inhibitor (GDI) and plotted column scatter graph. The error bars represent SEM. A.U. indicates arbitrary unit. $^*$p≤0.05, $^{**}$p≤0.01 or $^{***}$p≤0.001 compared with uninfected fat-unablated. $^#$p≤0.05, $^{##}$p≤0.01 or $^{###}$p≤0.001 compared with infected fat-unablated.

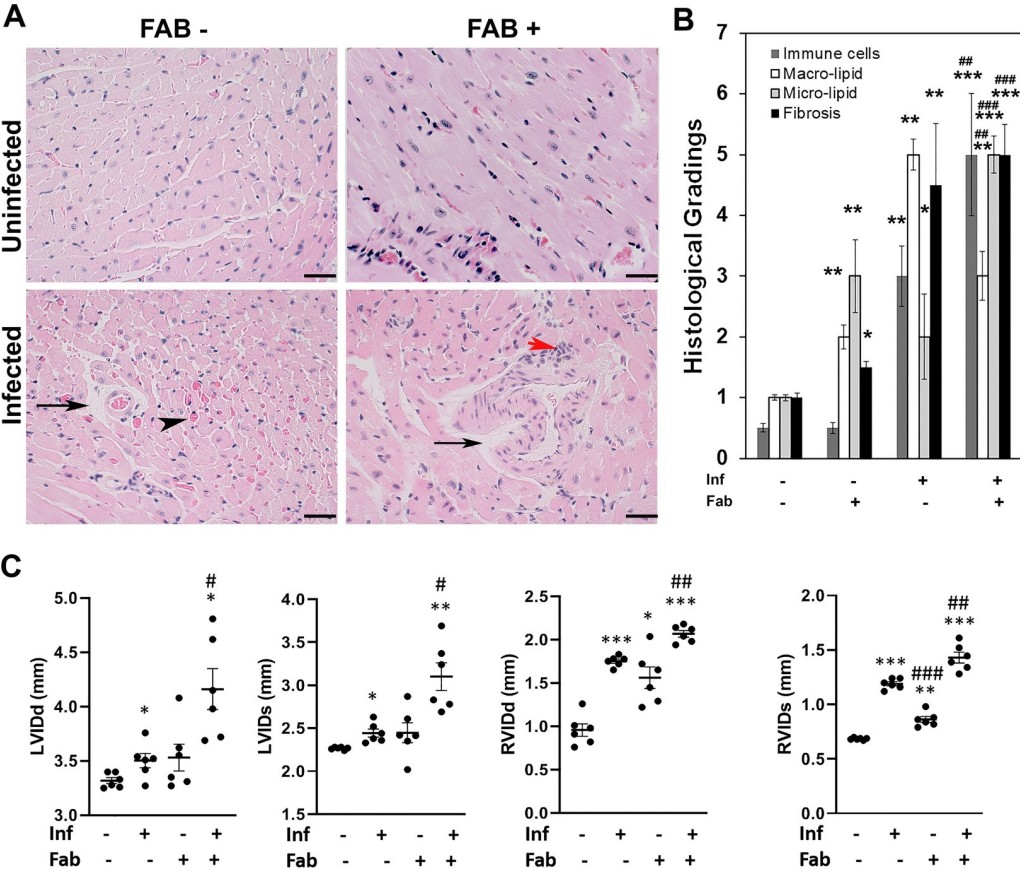

**Fig 8. Loss in fat cells exacerbates cardiac pathology and causes bi-ventricular enlargement in the hearts of chronic CD mice.** A. Hematoxylin and eosin (H&E) staining of hearts in indicated mice (infected or uninfected mice, fat-ablated (Fab +) or fat-unablated (Fab-)). Infiltrated immune cells, red arrowhead; vasculitis, black arrowhead; and presence of lipid droplets, black arrow. Bar = 100 μm, 20x magnification. B. Histologic grading of heart tissue pathology was carried out according to experimental groups and classified in terms of degree of infiltrated immune cells, size of adipocytes (macro lipid and micro lipid droplets), and fibrosis in the H&E sections of hearts in chronic *T. cruzi* infected mice with and without fat ablation (5 images per section/mouse in each group). Each class was graded on a six-point scale ranging from 0 to 5+ as discussed in Method section and presented as a bar graph. The values plotted are mean ± standard deviation (SD) from n = 5. C. Cardiac ultrasound imaging analysis in indicated mice (infected or uninfected mice, fat-ablated (Fab +) or fat-unablated (Fab-) 90DPI. Left ventricle internal diameter (LVID) and right ventricle internal diameter (RVID) both at diastole (d) and systole (s) conditions. The error bars represent SEM. A.U. indicates arbitrary unit. *p≤0.05, **p≤0.01 or ***p≤0.001 compared with uninfected fat-unablated. #p≤0.05, ##p≤0.01 or ###p≤0.001 compared with infected fat-unablated.

(both diastole and systole) in infected mice compared to control mice at 90DPI (Fig 8C). Fat ablation during the indeterminate stage in these infected mice significantly increased both LVID and RVID (diastole and systole) compared with both control and infected mice (Fig 8C). The LVID diastole and systole in infected fat-ablated mice were increased 1.1-fold (p<0.05) and 1.2-fold (p<0.05) respectively, compared with infected mice (Fig 8C). The increase in RVID diastole and systole in infected fat-ablated mice were 1.1-fold (p<0.01) and 1.1-fold (p<0.01) respectively, compared with infected mice (Fig 8C). Doppler flow profiles demonstrated a significant difference (p≤0.01) in the ejection fractions (%) between infected fat-ablated mice (EF%- 45 ± 4.2), and other groups; uninfected mice (EF%- 60±5.1), uninfected fat-ablated mice (EF%- 61 ± 4.7), and infected mice (EF%- 59 ± 6.4) (S3D Fig). The fractional shortening (FS) measurements showed a significant difference (p≤0.01) in the FS% between infected mice and infected fat-ablated mice which followed the similar trends of EF%

(S3D Fig). In other imaging studies it has been demonstrated that heart wall motion abnormalities and dilation of the RV are markers of Chagas disease severity in mice [3,4,6]. Decreased EF% and increased bi-ventricular dilation in infected fat-ablated mice display a severe form of cardiomyopathy associated with heart failure.

These data suggest that loss of fat cells during indeterminate stage increases cardiac parasite load, infiltration of immune cells and ER stress and the risk and progression of cardiomyopathy at the chronic stage of *T. cruzi* infection.

## Discussion

Acute myocarditis is mainly due to cardiac parasite load, infiltration of immune cells and inflammation leading to hypertrophic cardiomyopathy whereas, the pathogenesis and progression of chronic dilated cardiomyopathy is multifactorial. The clinical manifestations of chronic cardiomyopathy vary ranging from mild hypertrophic myocarditis to severe form of biventricular dilated cardiomyopathy. The effects of cardiac lipid accumulation on mitochondrial dysfunction and endoplasmic reticulum (ER) stress in the progression of dilated cardiomyopathy have been previously described [3,4,18]. In addition, it is known that mice infected with *T. cruzi* lose a significant amount of body fat and this may contribute to the pathogenesis of cardiomyopathy [16]. However, the specific role of adipocytes on cardiac lipid metabolism, mitochondrial and ER functions, inflammation and the pathogenesis of cardiomyopathy in chronic CD is currently not known.

To test the hypothesis that acute loss in fat cells increases the risk of cardiac pathogenesis, we used *T. cruzi* infected FAT-ATTAC mice in both acute and indeterminate/chronic murine CD models. Herein, we showed that ablation of fat cells during acute infection (mice infected with $5x10^4$ parasites) significantly increased the accumulation of micro-vesicular lipids, infiltration of immune cells, cardiac parasite load, led to an altered cardiac morphology (elevated LV and RV internal diameters) and reduced survival compared to infected mice without fat ablation (Figs 1 and 2).

The observation of reduced survival in fat ablation is in agreement with the previous studies where the authors showed that adipocytes/adipose tissue protected mice from acute *T. cruzi* infection, when mice were fed a high-fat diet, by acting as a reservoir for the parasites and sparing the heart from both parasites and inflammation [7]. During acute infection, a high-fat diet increased the amount of body fat in mice compared with mice fed a control diet [7], however, this high-fat diet aggravated the complications of cardiomyopathy in the chronic CD murine model [6]. Although, the chronic mice were fed a high-fat diet, they displayed a significant loss of body weight suggesting that loss in body weight may contribute to CCC [6]. Herein, we showed that ablation of fat cells during the indeterminate stage increases the risk of developing chronic cardiomyopathy and elevates severity of the disease which also agrees with clinical data showing significant loss in body weight and fat mass in patients with a severe form of CCC.

In our previous study, and confirmed by other authors, *T. cruzi* has been demonstrated to reside in adipose tissue and alter adipose tissue physiology during both acute and chronic infection in the murine CD model [16,19,20]. Adipocytes form a nutritional niche for parasites and an abrupt loss in adipocytes may trigger the translocation of parasites and stored lipids from adipose tissue into other organs including heart. Loss of adipocytes may trigger the infiltration of immune cells and induce pro-inflammatory signalling in adipose tissue [16]. In the current study, we demonstrated that fat-ablation during acute infection significantly increased parasite load and lipid droplets in the hearts of infected mice compared with the hearts of infected fat-unablated mice (Fig 1). Acute infection per se causes significant loss in adipocytes

and induces pro-inflammatory signalling in adipose tissue and alters adipose tissue physiology [20]. Although, the morphology and physiology of adipose tissue improves after the acute infection, due to the persistence of parasites in the adipose tissue, the adipocytes are compromised and not fully functional during the indeterminate/chronic stage.

Adipocyte physiology depends on the fine regulated balance between adipogenesis and lipolysis, which is deregulated during the indeterminate/chronic stages of infection. We demonstrated the presence of *T. cruzi* specific DNA in adipose tissue of chronically infected mice (90DPI) (S3A Fig). Our data suggest that persistence of parasites and infiltrated immune cells in adipose tissue alters adipose tissue physiology by causing an imbalance between adipogenesis and lipolysis. We observed the presence of disintegrated adipocytes with multi-ocular lipid droplets and infiltrated immune cells in adipose tissue at 90 DPI (Fig 3). The phenotype of adipocytes in the adipose tissue of infected mice was like pre-adipocytes as evident by increased levels of FABP4 and reduced levels of adipogenic markers of matured adipocytes such as adiponectin and PPARγ compared to control mice (Fig 4). However, fat ablation increased the levels of FABP4, adiponectin and PPARγ indicating that apoptotic cell death of adipocytes induces pro-adipogenic signaling. Adipose tissue from mice with chronic infection demonstrated a deregulated lipolysis pathway with decreased levels of HSL but increased levels of p-perilipin and PPARα. Perilipin is a regulatory protein that coats and protects the lipid droplets from lipolysis in adipocytes in the basal state [21]. Phosphorylation of perilipin exposes lipid droplets to HSL and activates lipolysis [22]. Although the levels of HSL and pHSL were lower in the adipose tissue of chronic infected mice compared to control mice, the ratio of pHSL/HSL was greater in infected mice compared with control mice. These data suggest that phosphorylated perilipin may be involved in the regulation of lipolysis and lipid oxidation as evident by pHSL/HSL ratio and the levels of PPARα, respectively (Fig 4). Fat ablation induced by apoptosis significantly elevated cardiac lipid droplets suggesting that the activated cell death pathways in adipose tissues of chronically infected mice may cause the chronic accumulation of lipid droplets in the hearts (Figs 5 and 8). Our data demonstrated that the presence of parasites increase infiltration of immune cells and induce adipocyte cell death via necrosis and apoptosis in adipose tissue during the indeterminate/early chronic infection in *T. cruzi* infected mice. These combined observations suggest that *T. cruzi* infection compromises adipocytes causing a leaky phenotype and any further shock to adipocytes during the indeterminate stage of infection, elevate lipolysis/loss of lipid droplets in adipose tissue and increase lipid accumulation in the hearts.

In previous studies, we and others had shown an accumulation of lipid droplets in the hearts of chronic CD mice and CD patients [17,23]. This accumulation of lipid droplets in the myocardium increases mitochondrial oxidative stress and ER stress, leads to mitochondrial dysfunction, cardiac cell death and fibrosis, and results in dilated cardiomyopathy. Interestingly, in the current study we found that an abundance of micro-vesicular lipid droplets in the hearts of infected fat-ablated mice and mostly macro-vesicular lipid droplets in the hearts of infected fat-unablated mice. These changes in lipid droplet sizes were associated with altered adipogenic signaling and lipid metabolism in the hearts of infected mice with and without fat ablation. Fat ablation significantly increased pre-adipocyte differentiation associated FABP4 and PPARγ signaling in the hearts and induced a breakdown of lipid droplets (causing micro-vesicular lipid droplets) as evident by increased levels of p-perilipin in the infected fat-ablated mice (Fig 6). Adipogenic adiponectin regulates PPARα [24]. Although the levels of fatty acid oxidation marker PPARα increases in the hearts, the catabolic pathways of fatty acid oxidation are impaired during the early chronic stage as shown by the decreased levels of ACSL, acetyl CoA carboxylase and ACAT1. These data suggest that the mitochondrial functioning is reduced during early chronic infection which is also supported by the data showing the

reduced levels of pyruvate dehydrogenase and cytochrome in the hearts of infected mice (Fig 7).

Cardiac lipidopathy induces ER stress due to increased lipid oxidation.[3] Our data also suggested increased ER stress in the hearts during chronic infection due to the accumulation of lipids as evidenced by the presence of significantly increased CHOP and BIP, and the levels of CHOP further elevated in the infected fat-ablated mice compared with infected fat-unablated mice (Fig 7). Fat ablation further increased lipid oxidation as evidenced by increased PPARα and SOD during infection which could have caused a further increase in cardiac ER stress.

We found that the levels of pro-inflammatory cytokines TNFα and IFNγ were significantly reduced in the hearts during chronic infection even though the levels of infiltrated immune cells (macrophages) were significantly higher compared to control mice (Fig 7). The reduction in TNFα may be due to the increased levels of adiponectin in the hearts of infected mice, which is known to regulate TNFα [25]. Fat-ablation further significantly increased the levels of macrophages in the hearts and decreased TNFα, however, interestingly increased IFNγ levels. These data suggest that degradation of macro-vesicular lipid droplets may activate IFNγ producing immune cells and induce cardiac cell death and fibrosis. We observed increased fibrosis in the hearts of infected fat-ablated mice, however, no significant difference was observed compared with the infected fat-unablated mice.

Our data establish a strong association between an increase in the loss of fat cells (adipose tissue) and cardiac adipogenesis and impairment of lipid metabolism. We also demonstrated a strong correlation between cardiac adipognesis/lipid metabolism and progression of cardiomyopathy [3]. Increased levels of loss of fat cells aggravated cardiac adipogenic/lipid metabolism and caused severe form of cardiomyopathy displaying biventricular enlargement (Fig 8). All together these observations suggest that *T. cruzi* infection compromised adipocytes/adipose tissue induced cardiac adipogenesis and impaired mitochondrial and ER functions may lead to cardiac cell death and fibrosis. This suggests that fat tissue has a significant role in the pathogenesis of cardiomyopathy and may be involved in the transition of the progression of the indeterminate stage of infection to chronic cardiomyopathy. Furthermore, a sudden loss of adipocytes/adipose tissue not only increases the risk of cardiomyopathy, but also worsens the severity of chronic CD.

Our results argue for a more potent role for adipocytes in regulating cardiac lipid metabolism, mitochondrial dysfunction, ER stress, inflammation and progression and severity of cardiomyopathy in Chronic CD. Considering the proximity of adipose tissue to and as a part of myocardium (epicardial and pericardial fat tissue), dysfunctional adipose tissue not only affects cardiac metabolism but also the inflammatory status, morphology and physiology of the myocardium. This association is highly important given that both the myocardium and adipose tissue are compromised in *T. cruzi* infection. In summary, our study ascribes a central role of adipocytes pathophysiology in the pathogenesis and severity of cardiomyopathy in CD. There is a direct correlation between an acute loss of body fat and the severe form of cardiomyopathy in infected mice. Further studies are warranted to understand the mechanism(s) of interactions of pathological myocardial fat with the myocardium during the transition of chronic CD.

## Supporting information

**S1 Fig.** Schematic representation of the experimental design of the acute (a) and chronic (b) induction of disease in mice.
(TIF)

**S2 Fig. Effect of fat ablation on survival of mice and cardiac functions in acute CD murine model.** A. Survival plot of acute *T. cruzi* infected mice with and without fat ablation. B. Change

in ejection fraction (EF) and fractional shortening (FS) analyzed by cardiac imaging. $^*$p$\leq$0.05, $^{**}$p$\leq$0.01 or $^{***}$p$\leq$0.001 compared with uninfected fat-unablated. $^\#$p$\leq$0.05, $^{\#\#}$p$\leq$0.01 or $^{\#\#\#}$p$\leq$0.001 compared with infected fat-unablated.
(TIF)

**S3 Fig. Effect of fat ablation on parasite persistence, cardiac morphology and functioning in chronic CD murine model (90DPI).** A. Assessment of the parasite load by quantitative PCR in adipose tissue of infected (Inf) and infected fat-ablated (Inf+FAB) mice, $^*$P$\leq$0.05. B. Assessment of the parasite load by quantitative PCR in the hearts of infected (Inf) and infected fat-ablated (Inf+FAB) mice, $^*$P$\leq$0.05. C. Hematoxylin and eosin (H&E) and masson trichrome staining of hearts in indicated mice (infected or uninfected mice, fat-ablated (Fab+) or fat-unablated (Fab-). Presence of lipid macro-lipid droplets (black arrow) and micro-lipid granules (black arrowhead). Bar = 100 µm, 20x magnification. D. Change in ejection fraction (EF) and fractional shortening (FS) analyzed by cardiac imaging. $^{**}$p$\leq$0.01 compared with uninfected fat-unablated.
(TIF)

# Acknowledgments

We thank members of the Rutgers Imaging Center and Histology Core Facility for their assistance with tissue collection, histological analysis, and image acquisition.

# Author Contributions

**Conceptualization:** Jyothi F. Nagajyothi.

**Data curation:** Kezia Lizardo, Janeesh P. Ayyappan, Louis M. Weiss, Philipp E. Scherer.

**Formal analysis:** Neelam Oswal, Jyothi F. Nagajyothi.

**Funding acquisition:** Jyothi F. Nagajyothi.

**Investigation:** Kezia Lizardo, Janeesh P. Ayyappan.

**Writing – original draft:** Neelam Oswal, Jyothi F. Nagajyothi.

**Writing – review & editing:** Louis M. Weiss, Jyothi F. Nagajyothi.

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
