## [Decision Letter · Decision Letter 0]

12 Feb 2021

Dear Dr Nagajyothi,

Thank you very much for submitting your manuscript "Fat Tissue Regulates the Pathogenesis and Severity of Cardiomyopathy in Murine Chagas Disease" for consideration at PLOS Neglected Tropical Diseases. As with all papers reviewed by the journal, your manuscript was reviewed by members of the editorial board and by several independent reviewers. The reviewers appreciated the attention to an important topic. Based on the reviews, we are likely to accept this manuscript for publication, providing that you modify the manuscript according to the review recommendations. 

Sincerely,

Albert Descoteaux, PhD

Associate Editor

Nilson Zanchin

Deputy Editor

Reviewer's Responses to Questions

**Key Review Criteria Required for Acceptance?**

**Methods**

-Are the objectives of the study clearly articulated with a clear testable hypothesis stated?

-Is the study design appropriate to address the stated objectives?

-Is the population clearly described and appropriate for the hypothesis being tested?

-Is the sample size sufficient to ensure adequate power to address the hypothesis being tested?

-Were correct statistical analysis used to support conclusions?

-Are there concerns about ethical or regulatory requirements being met?

Reviewer #1: The general objectives appear to be consistent with the results presented; however, some inconsistencies are noted in the introduction. The authors affirm that the purpose is to investigate the role of “altered adipocyte levels and physiology, using a fat-amendable transgenic murine FAT-ATTAC (fat apoptosis through targeted activation of caspase 8) model, on the regulation of cardiac parasite load, parasite persistence, inflammation, mitochondrial stress, ER stress, and CCC progression and severity, and survival during acute and chronic T. cruzi infection”. However, there is no data regarding parasite load or persistence. CCC severity parameters presented were histopathologic analysis and the internal ventricular diameter, but more functional studies are absent. Although the authors present survival data for the acute model, it is missing for the chronic model.

Reviewer #2: Well designed study. Sufficient details are provided for most of the methods. Authors are encouraged to provide more information on how macro/micro lipids are assessed/scored.

It was also help other researchers to reproduce protocols if authors will add further details on how they utilize H&E staining to score apoptosis, cell death, micro/macro-lipid etc.

**Results**

-Does the analysis presented match the analysis plan?

-Are the results clearly and completely presented?

-Are the figures (Tables, Images) of sufficient quality for clarity?

Reviewer #1: I find that supporting figures 2A and B have to be figure 1 in the main text, as they provide much more information that current figure 1 alone. Are the survival curves provided in figure 2C statistical different?

To facilitate readers analysis of figure 2, I suggest that the * and # meaning be included in the figure keys where it is missing.

I find that supporting figures 2A and B have to be figure 1 in the main text, as they provide much more information than current figure 1 alone. Are the survival curves provided in figure 2C statistically different?

To facilitate readers' analysis of figures, I suggest that the * and # meaning be included in the figure legends.

The results presented in figure 1 and supplementary are consistent with the overall objectives. However, the authors’ claim that there is “increased cardiac cell death, increased parasite load increased cardiomyopathy and increased mortality in T. cruzi infected mice” deserves further confirmation. Specifically, how was cardiac death determined? Or it arises from the histological appreciation? Supporting figure 2C suggests increased mortality in the FAB+, infected mice, but was it statistically different? The Internal Ventricular Diameter was the only cardiac parameter that showed significant changes?

During the chronic or indeterminate phase, the parasite load in adipose tissue was evaluated? How the adipocyte cell death was evaluated? Authors state in line 296 that “These results suggest that persistence of infiltrated immune cells in adipose tissue alters adipose tissue physiology by causing an imbalance between adipogenesis and lipolysis”. How can be concluded that immune cell infiltration is the direct consequence of the alterations of adipogenesis and lipolysis? Probably a most straightforward conclusion is that parasite presence (in adipose tissue or elsewhere) might be the driving force for this phenomenon.

Those questions are answered in further paragraphs. Thus I suggest to include the preliminary conclusions in lines 258 and 296 in the discussion section. The same applies to the statement in line 320. In this respect, The direct relationship between adipocyte dysfunction and immune cells' infiltration is not apparent since, in FAB+ mice, this indicator is not increased. The relationship between infection and the presence of cell death markers, especially necrosis, is clearer (however, it would be desirable to verify the parasitic load in adipose tissue to confirm this relationship). It is not ruled out that the ablation itself is the cause of the activation of caspase 3, since the mice submitted to this procedure also present elevated levels of cleaved caspase 3. 

If the authors performed doppler echocardiography, why do they only present the diameters of the chambers? What happened to, for example, the ejection fraction and other parameters of heart function? Were they affected enough to support the morphological alterations shown.

Finally, why not to show the evidence for the fibrosis involvement in the infected hearts, at least as par of a support figure.

Reviewer #2: Data are clearly presented with sufficient description and conclusions in general are justified by presented results.

1) Data on how authors confirmed fat ablation levels should be added in supp. fig 1

2) Sometimes authors make a general conclusion statement of results. Example: after describing results from Fig 1/2, supplement fig1/2, authors conclude changes in many parameters with infection and ablation. While figures present many data points in support of conclusion, cell death and cardiomyopathy are not discussed in data/results. If a decline in LV and increase in RV diameter is considered cardiomyopathy then it should be defined before conclusion. 

3) It will be nice if authors can substantiate H&E findings with oil red O or PAS

**Conclusions**

-Are the conclusions supported by the data presented?

-Are the limitations of analysis clearly described?

-Do the authors discuss how these data can be helpful to advance our understanding of the topic under study?

-Is public health relevance addressed?

Reviewer #1: The results are consistent with the overall purpose of the study, and although some observations can be included to strengthen the data provided, the overall conclusion may be appropriate, especially when the previous evidence points in the same direction. In general, it is important to comprehend the influence of the adipose tissue (where parasites can reside) on the chronic disease's overall outcome.

Reviewer #2: Yes. It is an important study describing the role of fat in Chagas pathology vs. protection. The use of an innovative mouse model is a major strength. 

Minor comment: Manuscript needs some editing. Just for example line 142, male is repeated twice. Line 145 Figure 1: should be Figure 1.

**Editorial and Data Presentation Modifications?**

Reviewer #1: Please see the previous setions

Reviewer #2: (No Response)

**Summary and General Comments**

Reviewer #1: The work presented herein is a well-planned study that demonstrates adipose tissue's influence on the development of chronic chagasic cardiomyopathy. Some methodological observations are referred to the determination of the parasitic load and persistence (which could be through analysis via qPCR of the presence of genetic material in cardiac and adipose tissues) or the inclusion of functional cardiac parameters that strengthen the morphological data. Other minor observations have been included in the preceding sections.

Reviewer #2: This is an important study describing the role of fat in Chagas pathology vs. protection. The use of an innovative mouse model is a major strength.

PLOS authors have the option to publish the peer review history of their article (what does this mean?). If published, this will include your full peer review and any attached files.

Reviewer #1: No

Reviewer #2: No
---

## [Editor Report · Decision Letter 1]

10 Mar 2021

Dear Dr Nagajyothi, 

We are pleased to inform you that your manuscript 'Fat Tissue Regulates the Pathogenesis and Severity of Cardiomyopathy in Murine Chagas Disease' has been provisionally accepted for publication in PLOS Neglected Tropical Diseases.

Best regards,

Albert Descoteaux, PhD

Associate Editor

Nilson Zanchin

Deputy Editor

---

## [Editor Report · Acceptance letter]

1 Apr 2021

Dear Dr. Nagajyothi,

We are delighted to inform you that your manuscript, "Fat Tissue Regulates the Pathogenesis and Severity of Cardiomyopathy in Murine Chagas Disease," has been formally accepted for publication in PLOS Neglected Tropical Diseases.

Best regards,

Shaden Kamhawi

co-Editor-in-Chief

Paul Brindley

co-Editor-in-Chief
